# Content-based Unrestricted Adversarial Attack

**Zhaoyu Chen**[1,2,*] **Bo Li**[2,†]**, Shuang Wu**[2]**, Kaixun Jiang**[1]**, Shouhong Ding**[2]**, Wenqiang Zhang**[1,3,†]

[1]Academy for Engineering and Technology, Fudan University
[2]Youtu Lab, Tencent     [3]School of Computer Science, Fudan University
{zhaoyuchen20, wqzhang}@fudan.edu.cn,
njumagiclibo@gmail.com, {calvinwu, ericshding}@tencent.com

## Abstract

Unrestricted adversarial attacks typically manipulate the semantic content of an image (*e.g.*, color or texture) to create adversarial examples that are both effective and photorealistic, demonstrating their ability to deceive human perception and deep neural networks with stealth and success. However, current works usually sacrifice unrestricted degrees and subjectively select some image content to guarantee the photorealism of unrestricted adversarial examples, which limits its attack performance. To ensure the photorealism of adversarial examples and boost attack performance, we propose a novel unrestricted attack framework called Content-based Unrestricted Adversarial Attack. By leveraging a low-dimensional manifold that represents natural images, we map the images onto the manifold and optimize them along its adversarial direction. Therefore, within this framework, we implement Adversarial Content Attack (ACA) based on Stable Diffusion and can generate high transferable unrestricted adversarial examples with various adversarial contents. Extensive experimentation and visualization demonstrate the efficacy of ACA, particularly in surpassing state-of-the-art attacks by an average of 13.3-50.4% and 16.8-48.0% in normally trained models and defense methods, respectively.

## 1 Introduction

Deep neural networks (DNNs) have significantly progressed in many tasks [18, 7]. However, with the rise of adversarial examples, the robustness of DNNs has been dramatically challenged [16]. Adversarial examples show the vulnerability of DNNs and expose security vulnerabilities in many security-sensitive applications. To avoid potential risks and further research the robustness of DNNs, it is of great value to expose as many "blind spots" of DNNs as possible at the current research stage.

Nowadays, various methods are proposed to generate adversarial examples [4, 5, 6]. To maintain human visual imperceptibility and images' photorealism, adversarial perturbations within the constraint of $l_p$ norm are generated by these adversarial attacks. However, it is well known that the adversarial examples generated under $l_p$ norm have obvious limitations: firstly, they are not ideal in terms of perceptual similarity and are still easily perceptible by humans [24, 23, 62]; secondly, these adversarial perturbations are not natural enough and have an inevitable domain shift with the noise in the natural world, resulting in the adversarial examples being different from the hard examples that appear in the real world [64]. In addition, current defense methods against $l_p$ norm adversarial examples overestimate their abilities, known as the Dunning-Kruger effect [28]. It can effectively defend against $l_p$ norm adversarial examples but is not robust enough when facing new and unknown attacks [25]. Therefore, unrestricted adversarial attacks are beginning to emerge, using unrestricted but natural changes to replace small $l_p$ norm perturbations, which are more practically meaningful.

---

*This work was done when Zhaoyu Chen was an intern at Youtu Lab, Tencent.
†indicates corresponding authors.

37th Conference on Neural Information Processing Systems (NeurIPS 2023).

Existing unrestricted adversarial attacks generate adversarial examples based on image content such as shape, texture, and color. Shape-based unrestricted attacks [56, 1] iteratively apply small deformations to the image through a gradient descent step. Then, texture-based unrestricted attacks [2, 40] are introduced, which manipulate an image's general attributes (texture or style) to generate adversarial examples. However, texture-based attacks result in unnatural results and have low adversarial transferability. Researchers then discover that manipulating pixel values along dimensions generates more natural adversarial examples, leading to the rise of color-based unrestricted attacks [20, 30, 2, 61, 47, 60]. Nonetheless, color-based unrestricted attacks tend to compromise flexibility in unconstrained settings to guarantee the photorealism of adversarial examples. They are achieved either through reliance on subjective intuition and objective metrics or by implementing minor modifications, thereby constraining their potential for adversarial transferability.

Considering the aforementioned reasons, we argue that an ideal unrestricted attack should meet three criteria: **i)** it needs to maintain human visual imperceptibility and the photorealism of the images; **ii)** the attack content should be diverse, allowing for unrestricted modifications of image contents such as texture and color, while ensuring semantic consistency; **iii)** the adversarial examples should have a high attack performance so that they can transfer between different models. However, there is still a substantial disparity between the current and ideal attacks.

To address this gap, we propose a novel unrestricted attack framework called **Content-based Unrestricted Adversarial Attack**. Firstly, we consider mapping images onto a low-dimensional manifold. This low-dimensional manifold is represented by a generative model and expressed as a latent space. This generative model is trained on millions of natural images, possessing two characteristics: **i)** sufficient capacity to ensure the photorealism of generated images; **ii)** well-alignment of image contents with latent space ensures a diversity of content. Subsequently, more generalized images can be generated by walking along the low-dimensional manifold. Optimizing the adversarial objective on this latent space allows us to achieve more diverse adversarial contents. In this paper, we propose **Adversarial Content Attack (ACA)** utilizing the diffusion model as a low-dimensional manifold. Specifically, we employ Image Latent Mapping (ILM) to map images onto the latent space, and utilize Adversarial Latent Optimization (ALO) to optimize the latents, thereby generating unrestricted adversarial examples with high transferability. In conclusion, our main contributions are:

• We propose a novel attack framework called Content-based Unrestricted Adversarial Attack, which utilizes high-capacity and well-aligned low-dimensional manifolds to generate adversarial examples that are more diverse and natural in content.

• We achieve an unrestricted content attack, known as the Adversarial Content Attack. By utilizing Image Latent Mapping and Adversarial Latent Optimization techniques, we optimize latents in a diffusion model, generating high transferable unrestricted adversarial examples.

• The effectiveness of our attack has been validated through experimentation and visualization. Notably, we have achieved a significant improvement of **13.3∼50.4%** over state-of-the-art attacks in terms of adversarial transferability.

## 2 Background and Preliminary

**Problem Definition.** For a deep learning classifier $\mathcal{F}_\theta(\cdot)$ with parameters $\theta$, we denote the clean image as $x$ and the corresponding true label as $y$. Formally, unrestricted adversarial attacks aim to create imperceptible adversarial perturbations (such as image distortions, texture or color modifications, etc.) for a given input $x$ to generate an adversarial example $x_{adv}$ that can mislead the classifier $\mathcal{F}_\theta(\cdot)$:

$$\max_{x_{adv}} \mathcal{L}(\mathcal{F}_\theta(x_{adv}), y), \quad s.t.\ x_{adv} \text{ is natural}, \tag{1}$$

where $\mathcal{L}(\cdot)$ is the loss function. Because existing unrestricted attacks are limited by their attack contents, it prevents them from generating sufficiently natural adversarial examples and restricts their attack performance on different models. We hope that unrestricted adversarial examples are more natural and possess higher transferability. Therefore, we consider a more challenging and practical black-box setting to evaluate the attack performance of unrestricted adversarial examples. In contrast to the white-box setting, the black-box setting has no access to any information about the target model (i.e., architectures, network weights, and gradients). It can only generate adversarial examples by using a substitute model $\mathcal{F}_\phi(\cdot)$ and exploiting their transferability to fool the target model $\mathcal{F}_\theta(\cdot)$.

**Content-based Unrestricted Adversarial Attack.** Existing unconstrained attacks tend to modify fixed content in images, such as textures or colors, compromising flexibility in unconstrained settings to ensure the photorealism of adversarial examples. For instance, ColorFool [47] manually selects the parts of the image that are sensitive to human perception in order to modify the colors, and its effectiveness is greatly influenced by human intuition. Natural Color Fool [60], utilizes ADE20K [63] to construct the distribution of color distributions, resulting in its performance being restricted by the selected dataset. These methods subjectively choose the content to be modified for minor modifications, but this sacrifices the flexibility under unrestricted settings and limits the emergence of more "unrestricted" attacks. Taking this into consideration, we contemplate whether it is possible to achieve an unrestricted adversarial attack that can adaptively modify the content of an image while ensuring the semantic consistency of the image.

An ideal unrestricted attack should ensure the photorealism of adversarial examples, possess diverse adversarial contents, and exhibit potential attack performance. To address this gap, we propose a novel unrestricted attack framework called **Content-based Unrestricted Adversarial Attack**. Within this framework, we assume that natural images can be mapped onto a low-dimensional manifold by a generative model. As this low-dimensional manifold is well-trained on natural images, it naturally ensures the photorealism of the images and possesses the rich content present in natural images. Once we map an image onto a low-dimensional manifold, moving it along the adversarial direction on the manifold yields an unrestricted adversarial example. We argue that such a framework is closer to the ideal of unrestricted adversarial attacks, as it inherently guarantees the photorealism of adversarial examples and rich image content. Moreover, since the classifier itself fits the distribution of this low-dimensional manifold, adversarial examples generated along the manifold have more adversarial contents and the potential for strong attack performance, as shown in Figure 1.

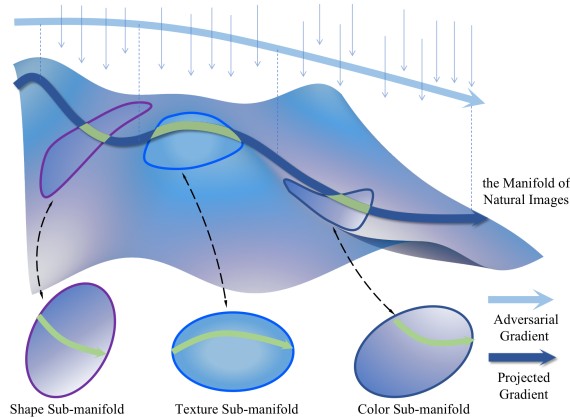

Figure 1: Adversarial examples are generated along the adversarial direction of the low-dimensional manifold of natural images. This manifold represents many contents of natural images, so the generated unrestricted adversarial examples combine multiple adversarial contents (shape, texture and color).

Naturally, selecting a low-dimensional manifold represented by a generative model necessitates careful consideration. There are two characteristics taken into account: **i)** sufficient capacity to ensure photorealism in the generated images; and **ii)** well-alignment ensures that image attributes are aligned with the latent space, thereby promoting diversity in content generation. Recently, diffusion models have emerged as a leading approach for generating high-quality images across varied datasets, frequently outperforming GANs [9]. However, several large-scale text-to-image diffusion models, including Imagen [44], DALL-E2 [41], and Stable Diffusion [42], have only recently come to the fore, exhibiting unparalleled semantic generation capabilities. Considering the trade-off between computational cost and high-fidelity image generation, we select Stable Diffusion as the low-dimensional manifold in this paper. It is based on prompt input and is capable of generating highly realistic natural images that conform to the semantics of the prompts.

## 3   Adversarial Content Attack

Based on the aforementioned framework and the full utilization of the diffusion model's capability, we achieve the unrestricted content-based attack known as **Adversarial Content Attack (ACA)**, as shown in Figure 2. Specifically, we first employ **Image Latent Mapping (ILM)** to map images onto the latent space represented by this low-dimensional manifold. Subsequently, we introduce an **Adversarial Latent Optimization (ALO)** technique that moves the latent representations of images along the adversarial direction on the manifold. Finally, based on iterative optimization, ACA can generate highly transferable unrestricted adversarial examples that appear quite natural. The

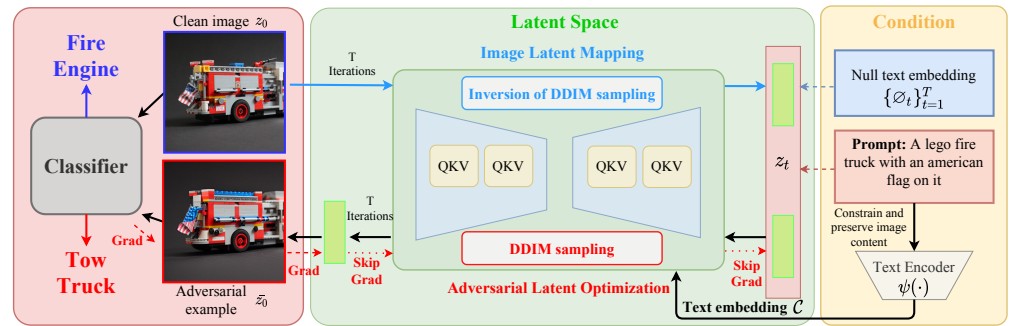

Figure 2: Pipeline of Adversarial Content Attack. First, we use Image Latent Mapping to map images into latent space. Next, Adversarial Latent Optimization is used to generate adversarial examples. Eventually, the generated adversarial examples can fool the target classifier.

algorithm for ACA is presented in Algorithm 1, and we further combine the diffusion model to design the corresponding mapping and optimization methods.

## 3.1 Image Latent Mapping

For the diffusion model, the easiest image mapping is the inverse process of DDIM sampling [9, 48] with the condition embedding $\mathcal{C} = \psi(\mathcal{P})$ of prompts $\mathcal{P}$, based on the assumption that the ordinary differential equation (ODE) process can be reversed in the limit of small steps:

$$z_{t+1} = \sqrt{\frac{\alpha_{t+1}}{\alpha_t}} z_t + \sqrt{\alpha_{t+1}} \left( \sqrt{\frac{1}{\alpha_{t+1}} - 1} - \sqrt{\frac{1}{\alpha_t} - 1} \right) \cdot \epsilon_\theta(z_t, t, \mathcal{C}), \tag{2}$$

where $z_0$ is the given real image, a schedule $\{\beta_0, ..., \beta_T\} \in (0, 1)$ and $\alpha_t = \prod_1^t (1 - \beta_i)$. In general, this process is the reverse direction of the denoising process ($z_0 \rightarrow z_T$ instead of $z_T \rightarrow z_0$), which can map the image $z_0$ to $z_T$ in the latent space. Image prompts are automatically generated using image caption models (e.g., BLIP v2 [31]). For simplicity, the encoding of the VAE is ignored.

Text-to-image synthesis usually emphasizes the effect of the prompt. Therefore, a classifier-free guidance technique [19] is proposed. Its prediction is also performed unconditionally, which is then extrapolated with the conditioned prediction. Given $w$ as the guidance scale parameter and $\varnothing = \psi("")$ as the embedding of a null text, the classifier-free guidance prediction is expressed by:

$$\tilde{\epsilon}_\theta(z_t, t, \mathcal{C}, \varnothing) = w \cdot \epsilon_\theta(z_t, t, \mathcal{C}) + (1 - w) \cdot \epsilon_\theta(z_t, t, \varnothing), \tag{3}$$

where $w = 7.5$ is the default value for Stable Diffusion. However, since the noise is predicted by the model $\epsilon_\theta$ in the inverse process of DDIM sampling, a slight error is incorporated in every step. Due to the existence of a large guidance scale parameter $w$ in the classifier-free guidance technique, slight errors are amplified and lead to cumulative errors. Consequently, executing the inverse process of DDIM sampling with classifier-free guidance not only disrupts the Gaussian distribution of noises but also induces visual artifacts of unreality [37].

To mitigate cumulative errors, we follow [37] and optimize a null text embedding $\varnothing_t$ for each timestamp $t$. First, the inverse process of DDIM sampling with $w = 1$ outputs a series of consecutive latents $\{z_0^*, ..., z_T^*\}$ where $z_0^* = z_0$. Then, we conduct the following optimization with $w = 7.5$ (the default value for Stable Diffusion) and $\bar{z}_T = z_t$ during $N$ iterations for the timestamps $t = \{T, ..., 1\}$:

$$\min_{\varnothing_t} ||z_{t-1}^* - z_{t-1}(\bar{z}_t, t, \mathcal{C}, \varnothing_t)||_2^2, \tag{4}$$

$$z_{t-1}(\bar{z}_t, t, \mathcal{C}, \varnothing_t) = \sqrt{\frac{\alpha_{t-1}}{\alpha_t}} \bar{z}_t + \sqrt{\alpha_{t-1}} \left( \sqrt{\frac{1}{\alpha_{t-1}} - 1} - \sqrt{\frac{1}{\alpha_t} - 1} \right) \cdot \tilde{\epsilon}_\theta(z_t, t, \mathcal{C}, \varnothing_t). \tag{5}$$

At the end of each step, we update $z_{t-1}^-$ to $z_{t-1}(\bar{z}_t, t, \mathcal{C}, \varnothing_t)$. Finally, we obtain the latent of the given image in the low-dimensional manifold, consisting of the noise $\bar{z}_T$, the null text embedding $\varnothing_t$, and the text embedding $\mathcal{C} = \psi(\mathcal{P})$. Note that compared to other strategies [12, 43, 26, 54], the current strategy is simple and effective and does not require fine-tuning to obtain high-quality image reconstruction. Next, we exploit this latent to generate unrestricted adversarial examples.

## 3.2 Adversarial Latent Optimization

In this section, we propose an optimization method for latents to maximize the attack performance on unrestricted adversarial examples. In the latent space of a given image after ILM, the null text embedding $\varnothing_t$ ensures the quality of the reconstructed image, while the text embedding $\mathcal{C}$ ensures the semantic information of the image. Therefore, optimizing both embeddings may not be ideal. Considering that the noise $\bar{z}_T$ largely represents the image's information in the latent space, we choose to optimize it instead. However, this optimization is still challenged by complex gradient calculations and the overflow of the value range.

Based on the latents generated by ILM, we define the denoising process of diffusion models as $\Omega(\cdot)$ through Equation 5, and it involves $T$ iterations:

$$\Omega(z_T, T, \mathcal{C}, \{\varnothing_t\}_{t=1}^T) = z_0\left(z_1\left(..., (z_{T-1}, T-1, \mathcal{C}, \varnothing_{T-1}), ..., 1, \mathcal{C}, \varnothing_1\right), 0, \mathcal{C}, \varnothing_0\right). \quad (6)$$

Therefore, the reconstructed image is denoted as $\bar{z}_0 = \Omega(z_T, T, \mathcal{C}, \{\varnothing_t\})$. The computational process of VAE is disregarded herein, as it is differentiable. Combining Equation 7, our adversarial objective optimization is expressed by:

$$\max_{\delta} \mathcal{L}\left(\mathcal{F}_\theta(\bar{z}_0), y\right), \quad s.t. \ ||\delta||_\infty \leq \kappa, \bar{z}_0 = \Omega(z_T + \delta, T, \mathcal{C}, \{\varnothing_t\}) \text{ and } \bar{z}_0 \text{ is natural}, \quad (7)$$

where $\delta$ is the adversarial perturbation on the latent space. Our loss function consists of two parts: **i)** cross-entropy loss $\mathcal{L}_{ce}$, which mainly guides adversarial examples toward misclassification. **ii)** mean square error loss $\mathcal{L}_{mse}$ mainly guides the generated adversarial examples to be as close as possible to clean images on $l_2$ distance. Therefore, the total loss function $\mathcal{L}$ is expressed as:

$$\mathcal{L}(\mathcal{F}_\theta(\bar{z}_0), y, z_0) = \mathcal{L}_{ce}(\mathcal{F}_\theta(\bar{z}_0), y)) - \beta \cdot \mathcal{L}_{mse}(\bar{z}_0, z_0), \quad (8)$$

where $\beta$ is 0.1 in this paper. The loss function $\mathcal{L}$ aims to maximize the cross-entropy loss and minimize the $l_2$ distance between the adversarial example $\bar{z}_0$ and the clean image $z_0$.

To ensure the consistency of $z_0$ and $\bar{z}_0$, we assume that $\delta$ does not change the consistency when $\delta$ is extremely small, *i.e.*, $||\delta||_\infty \leq \kappa$. The crux pertains to determining the optimal $\delta$ that yields the maximum classification loss. Analogous to conventional adversarial attacks, we employ gradient-based techniques to estimate $\delta$ through: $\delta \simeq \eta \nabla_{z_T} \mathcal{L}\left(\mathcal{F}_\theta(\bar{z}_0), y\right)$, where $\eta$ denotes the magnitude of perturbations that occur in the direction of the gradient. To expand $\nabla_{z_T} \mathcal{L}\left(\mathcal{F}_\theta(\bar{z}_0), y\right)$ by the chain rule, we can have these derivative terms as follows:

$$\nabla_{z_T} \mathcal{L}\left(\mathcal{F}_\theta(\bar{z}_0), y\right) = \frac{\partial \mathcal{L}}{\partial \bar{z}_0} \cdot \frac{\partial \bar{z}_0}{\partial z_1} \cdot \frac{\partial z_1}{\partial z_2} \cdots \frac{\partial z_{T-1}}{\partial z_T}. \quad (9)$$

**Skip Gradient.** After observing the items, we find that although each item is differentiable, it is not feasible to derive the entire calculation graph. First, we analyze the term $\frac{\partial \mathcal{L}}{\partial \bar{z}_0}$, which represents the derivative of the classifier with respect to the reconstructed image $\bar{z}_0$ and provides the adversarial gradient direction. Then, for $\frac{\partial z_t}{\partial z_{t+1}}$, each calculation of the derivative represents the calculation of a backpropagation. Furthermore, a complete denoising process accumulates $T$ calculation graphs, resulting in memory overflow (similar phenomena are also found in [45]). Therefore, the gradient of the denoising process cannot be directly calculated.

Fortunately, we propose a skip gradient to approximate $\frac{\partial z_0}{\partial z_T} = \frac{\partial \bar{z}_0}{\partial z_1} \cdot \frac{\partial z_1}{\partial z_2} \cdots \frac{\partial z_{T-1}}{\partial z_T}$. Recalling the diffusion process, the denoising process aims to eliminate the Gaussian noise added in DDIM sampling [48, 9, 42]. DDIM samples $z_t$ at any arbitrary time step $t$ in a closed form using reparameterization trick:

$$z_t = \sqrt{\alpha_t} z_0 + \sqrt{1 - \alpha_t} \varepsilon, \quad \varepsilon \sim \mathcal{N}(0, I). \quad (10)$$

Consequently, we perform a manipulation by rearranging Equation 10 to obtain $z_0 = \frac{1}{\sqrt{\alpha_t}} z_t - \sqrt{\frac{1-\alpha_t}{\alpha_t}} \varepsilon$. Hence, we further obtain $\frac{\partial z_0}{\partial z_t} = \frac{1}{\sqrt{\alpha_t}}$. In Stable Diffusion, timestep $t$ is at most 1000, so $\lim_{t \to 1000} \frac{\partial z_0}{\partial z_t} = \lim_{t \to 1000} \frac{1}{\sqrt{\alpha_t}} \approx 14.58$. In summary, $\frac{\partial z_0}{\partial z_t}$ can be regarded as a constant $\rho$ and Equation 9 can be re-expressed as $\nabla_{z_T} \mathcal{L}\left(\mathcal{F}_\theta(\bar{z}_0), y\right) = \rho \frac{\partial \mathcal{L}}{\partial \bar{z}_0}$. In summary, skip gradients approximate the gradients of the denoising process while reducing the computation and memory usage.

**Differentiable Boundary Processing.** Since the diffusion model does not explicitly constrain the value range of $\bar{z}_0$, the modification of $z_T$ may cause the value range to be exceeded. So we introduce

**Algorithm 1** Adversarial Content Attack

**Input:** a input image $z_0$ with the label $y$, a text embedding $\mathcal{C} = \psi(\mathcal{P})$, a classifier $\mathcal{F}_\theta(\cdot)$, DDIM steps $T$, image mapping iteration $N_i$, attack iterations $N_a$, and momentum factor $\mu$
1: Calculate latents $\{z_0^*, ..., z_T^*\}$ using Equation 5 over $z_0$ with $w = 1$
2: Initialize $w = 7.5, \bar{z}_T \leftarrow z_T^*, \varnothing \leftarrow \psi("")$, $\delta_0 \leftarrow 0, g_0 \leftarrow 0$
3: $//$ *Image Latent Mapping*
4: **for** $t = T, T-1 \ldots, 1$ **do**
5:     **for** $j = 1, \ldots, N_i$ **do**
6:         $\varnothing_t \leftarrow \varnothing_t - \zeta \nabla_{\varnothing_t} ||z_{t-1}^* - z_{t-1}(\bar{z}_t, t, \mathcal{C}, \varnothing_t)||_2^2$
7:     **end for**
8:     $z_{t-1}^- \leftarrow z_{t-1}(\bar{z}_t, t, \mathcal{C}, \varnothing_t), \varnothing_{t-1} \leftarrow \varnothing_t$
9: **end for**
10: $//$ *Adversarial Latent Optimization*
11: **for** $k = 1, \ldots, N_a$ **do**
12:     $\bar{z}_0 \leftarrow \Omega\left(\bar{z}_T + \delta_{k-1}, T, \mathcal{C}, \{\varnothing_t\}_{t=1}^T\right)$
13:     $g_k \leftarrow \mu \cdot g_{k-1} + \frac{\nabla_{z_T}\mathcal{L}(\mathcal{F}_\theta(\varrho(\bar{z}_0), y))}{||\nabla_{z_T}\mathcal{L}(\mathcal{F}_\theta(\varrho(\bar{z}_0), y))||_1}$
14:     $\delta_k \leftarrow \Pi_\kappa\left(\delta_{k-1} + \eta \cdot \text{sign}(g_k)\right)$
15: **end for**
16: $\bar{z}_0 \leftarrow \varrho\left(\Omega\left(\bar{z}_T + \delta_{N_a}, T, \mathcal{C}, \{\varnothing_t\}_{t=1}^T\right)\right)$
**Output:** The unrestricted adversarial example $\bar{z}_0$.

differentiable boundary processing $\varrho(\cdot)$ to solve this problem. $\varrho(\cdot)$ constrains the values outside $[0, 1]$ to the range of $[0, 1]$. The mathematical expression of DPB is as follows:

$$\varrho(x) = \begin{cases} \tanh(1000x)/10000, & x < 0, \\ x, & 0 \le x \le 1, \\ \tanh(1000(x-1))/10001, & x > 1. \end{cases} \quad (11)$$

Next, we define $\Pi_\kappa$ as the projection of the adversarial perturbation $\delta$ onto $\kappa$-ball. We introduce momentum $g$ and express the optimization adversarial latents as:

$$g_k \leftarrow \mu \cdot g_{k-1} + \frac{\nabla_{z_T}\mathcal{L}\left(\mathcal{F}_\theta\left((\varrho(\bar{z}_0), y)\right)\right)}{||\nabla_{z_T}\mathcal{L}\left(\mathcal{F}_\theta\left(\varrho(\bar{z}_0), y)\right)\right)||_1}, \quad (12)$$

$$\delta_k \leftarrow \Pi_\kappa\left(\delta_{k-1} + \eta \cdot \text{sign}(g_k)\right). \quad (13)$$

In general, Adversarial Latent Optimization (ALO) employs skip gradient to determine the gradient of the denoising process, and integrates differentiable boundary processing to regulate the value range of adversarial examples, and finally performs iterative optimization according to the gradient. Combined with Image Latent Mapping, Adversarial Content Attack is illustrated in Algorithm 1.

## 4 Experiments

### 4.1 Experimental Setup

**Datasets.** Our experiments are conducted on the ImageNet-compatible Dataset [29]. The dataset consists of 1,000 images from ImageNet's validation set [8], and is widely used in [10, 13, 58, 60].

**Attack Evaluation.** We choose SAE [20], ADer [1], ReColorAdv [30], cAdv [2], tAdv [2], ACE [61], ColorFool [47], NCF [60] as comparison methods of Adversarial Content Attack (ACA). The parameters for these unrestricted attacks follow the corresponding default settings. Our attack evaluation metric is the attack success rate (ASR, %), which is the percentage of misclassified images.

**Models.** To evaluate the adversarial robustness of network architectures, we select convolutional neural networks (CNNs) and vision transformers (ViTs) as the attacked models, respectively. For CNNs, we choose normally trained MoblieNet-V2 (MN-v2) [46], Inception-v3 (Inc-v3) [50], ResNet-50 (RN-50) and ResNet-152 (RN-152) [18], Densenet-161 (Dense-161) [22], and EfficientNet-b7 (EF-b7) [52]. For ViTs, we consider normally trained MoblieViT (MobViT-s) [35], Vision Transformer (ViT-B) [11], Swin Transformer (Swin-B) [34], and Pyramid Vision Transformer (PVT-v2) [55].

Table 1: Performance comparison of adversarial transferability on normally trained CNNs and ViTs. We report attack success rates (%) of each method ("*" means white-box attack results).

| Surrogate Model | Attack | Models | | | | | | | | | | Avg. ASR (%) |
|---|---|---|---|---|---|---|---|---|---|---|---|---|
| | | CNNs | | | | | | Transformers | | | | |
| | | MN-v2 | Inc-v3 | RN-50 | Dense-161 | RN-152 | EF-b7 | MobViT-s | ViT-B | Swin-B | PVT-v2 | |
| - | Clean | 12.1 | 4.8 | 7.0 | 6.3 | 5.6 | 8.7 | 7.8 | 8.9 | 3.5 | 3.6 | 6.83 |
| | ILM | 13.5 | 5.5 | 8.0 | 6.3 | 5.9 | 8.3 | 8.3 | 9.0 | 4.8 | 4.0 | 7.36 |
| MobViT-s | SAE | 60.2 | 21.2 | 54.6 | 42.7 | 44.9 | 30.2 | 82.5* | 38.6 | 21.1 | 20.2 | 37.08 |
| | ADef | 14.5 | 6.6 | 9.0 | 8.0 | 7.1 | 9.8 | 80.8* | 9.7 | 5.1 | 4.6 | 8.27 |
| | ReColorAdv | 37.4 | 14.7 | 26.7 | 22.4 | 21.0 | 20.8 | 96.1* | 21.5 | 16.3 | 16.7 | 21.94 |
| | cAdv | 41.9 | 25.4 | 33.2 | 31.2 | 28.2 | 34.7 | 84.3* | 32.6 | 22.7 | 22.0 | 30.21 |
| | tAdv | 33.6 | 18.8 | 22.1 | 18.7 | 18.7 | 15.8 | 97.4* | 15.3 | 11.2 | 13.7 | 18.66 |
| | ACE | 30.7 | 9.7 | 20.3 | 16.3 | 14.4 | 13.8 | 99.2* | 16.5 | 6.8 | 5.8 | 14.92 |
| | ColorFool | 47.1 | 12.0 | 40.0 | 28.1 | 30.7 | 19.3 | 81.7* | 24.3 | 9.7 | 10.0 | 24.58 |
| | NCF | **67.7** | 31.2 | 60.3 | 41.8 | 52.2 | 32.2 | 74.5* | 39.1 | 20.8 | 23.1 | 40.93 |
| | ACA (Ours) | 66.2 | **56.6** | **60.6** | **58.1** | **55.9** | **55.5** | 89.8* | **51.4** | **52.7** | **55.1** | **56.90** |
| MN-v2 | SAE | 90.8* | 22.5 | 53.2 | 38.0 | 41.9 | 26.9 | 44.6 | 33.6 | 16.8 | 18.3 | 32.87 |
| | ADer | 56.6* | 7.6 | 8.4 | 7.7 | 7.1 | 10.9 | 11.7 | 9.5 | 4.5 | 4.5 | 7.99 |
| | ReColorAdv | 97.7* | 18.6 | 33.7 | 24.7 | 26.4 | 20.7 | 31.8 | 17.7 | 12.2 | 12.6 | 22.04 |
| | cAdv | 96.6* | 26.8 | 39.6 | 33.9 | 29.9 | 32.7 | 41.9 | 33.1 | 20.6 | 19.7 | 30.91 |
| | tAdv | 99.9* | 27.2 | 31.5 | 24.3 | 24.5 | 22.4 | 40.5 | 16.1 | 15.9 | 15.1 | 24.17 |
| | ACE | 99.1* | 9.5 | 17.9 | 12.4 | 12.6 | 11.7 | 16.3 | 12.1 | 5.4 | 5.6 | 11.50 |
| | ColorFool | 93.3* | 9.5 | 25.7 | 15.3 | 15.4 | 13.4 | 15.7 | 14.2 | 5.9 | 6.4 | 13.50 |
| | NCF | 93.2* | 33.6 | **65.9** | 43.5 | **56.3** | 33.0 | 52.6 | 35.8 | 21.2 | 20.6 | 40.28 |
| | ACA (Ours) | 93.1* | **56.8** | 62.6 | **55.7** | 56.0 | **51.0** | 59.6 | 48.7 | 48.6 | 50.4 | **54.38** |
| RN-50 | SAE | 63.2 | 25.9 | 88.0* | 41.9 | 46.5 | 28.8 | 45.9 | 35.3 | 20.3 | 19.6 | 36.38 |
| | ADer | 15.5 | 7.7 | 55.7* | 8.4 | 7.8 | 11.4 | 12.3 | 9.2 | 4.4 | 4.9 | 9.09 |
| | ReColorAdv | 40.6 | 17.7 | 96.4* | 28.3 | 33.3 | 19.2 | 29.3 | 18.8 | 12.9 | 13.4 | 23.72 |
| | cAdv | 44.2 | 25.3 | 97.2* | 36.8 | 37.0 | 34.9 | 40.1 | 30.6 | 19.3 | 20.2 | 32.04 |
| | tAdv | 43.4 | 27.0 | 99.0* | 28.8 | 30.2 | 21.6 | 35.9 | 16.5 | 15.2 | 15.1 | 25.97 |
| | ACE | 32.8 | 9.4 | 99.1* | 16.1 | 15.2 | 12.7 | 20.5 | 13.1 | 6.1 | 5.3 | 14.58 |
| | ColorFool | 41.6 | 9.8 | 90.1* | 18.6 | 21.0 | 15.4 | 20.4 | 15.4 | 5.9 | 6.8 | 17.21 |
| | NCF | **71.2** | 33.6 | 91.4* | 48.5 | 60.5 | 32.4 | 52.6 | 36.8 | 19.8 | 21.7 | 41.90 |
| | ACA (Ours) | 69.3 | **61.6** | 88.3* | **61.9** | **61.7** | **60.3** | 62.6 | 52.9 | 51.9 | 53.2 | **59.49** |
| ViT-B | SAE | 54.5 | 26.9 | 49.7 | 38.4 | 41.4 | 30.4 | 46.1 | 78.4* | 19.9 | 18.1 | 36.16 |
| | ADer | 15.3 | 8.3 | 9.9 | 8.4 | 7.6 | 12.0 | 12.4 | 81.5* | 5.3 | 5.5 | 9.41 |
| | ReColorAdv | 25.5 | 12.1 | 17.5 | 13.9 | 14.4 | 15.4 | 22.9 | 97.7* | 10.9 | 8.6 | 15.69 |
| | cAdv | 31.4 | 27.0 | 26.1 | 22.5 | 19.9 | 26.1 | 32.9 | 96.5* | 18.4 | 16.9 | 24.58 |
| | tAdv | 39.5 | 22.8 | 25.8 | 23.2 | 22.3 | 20.8 | 34.1 | 93.5* | 16.3 | 15.3 | 24.46 |
| | ACE | 30.9 | 11.4 | 22.0 | 15.5 | 15.2 | 13.0 | 17.0 | 98.6* | 6.5 | 6.3 | 15.31 |
| | ColorFool | 45.3 | 13.9 | 35.7 | 24.3 | 28.8 | 19.8 | 27.0 | 83.1* | 8.9 | 9.3 | 23.67 |
| | NCF | 55.9 | 25.3 | 50.6 | 34.8 | 42.3 | 29.9 | 40.6 | 81.0* | 20.0 | 19.1 | 35.39 |
| | ACA (Ours) | **64.6** | **58.8** | **60.2** | **58.1** | **58.1** | **57.1** | **60.8** | 87.7* | **55.5** | **54.9** | **58.68** |

**Implementation Details.** Our experiments are run on an NVIDIA Tesla A100 with Pytorch. DDIM steps $T = 50$, image mapping iteration $N_i = 10$, attack iterations $N_a = 10$, $\beta = 0.1$, $\zeta = 0.01$, $\eta = 0.04$, $\kappa = 0.1$, and $\mu = 1$. The version of Stable Diffusion [42] is v1.4. Prompts for images are automatically generated using BLIP v2 [31].

## 4.2 Attacks on Normally Trained Models

In this section, we assess the adversarial transferability of normally trained convolutional neural networks (CNNs) and vision transformers (ViTs), including methods such as SAE [20], ADer [1], ReColorAdv [30], cAdv [2], tAdv [2], ACE [61], ColorFool [47], NCF [60], and our ACA. Adversarial examples are crafted via MobViT-s, MN-v2, RN-50, and ViT-B, respectively. Avg. ASR (%) refers to the average attack success rate on non-substitute models.

Table 1 illustrates the performance comparison of adversarial transferability on normally trained CNNs and ViTs. It can be observed that adversarial examples by ours generally exhibit superior transferability compared to those generated by state-of-the-art competitors and the impact of ILM on ASR is exceedingly marginal. When CNNs (RN-50 and MN-v2) are used as surrogate models, our ACA exhibits minimal differences with state-of-the-art NCF in MN-v2, RN-50, and RN-152. However, in Inc-v3, Dense-161, and EF-b7, such as when RN-50 is used as the surrogate model, we significantly outperform NCF by **28.0%**, **13.4%** and **27.9%**, respectively. This indicates that our ACA has higher transferability in heterogeneous CNNs. Furthermore, our ACA demonstrates state-of-the-art transferability in current unconstrained attacks under the more challenging cross-architecture

Table 2: Performance comparison of adversarial transferability on adversarial defense methods.

| Attack | HGD | R&P | NIPS-r3 | JPEG | Bit-Red | DiffPure | Inc-v3$_{ens3}$ | Inc-v3$_{ens4}$ | IncRes-v2$_{ens}$ | Res-De | Shape-Res | Avg. ASR (%) |
|--------|-----|-----|---------|------|---------|----------|-----------------|-----------------|-------------------|--------|-----------|--------------|
| Clean | 1.2 | 1.8 | 3.2 | 6.2 | 17.6 | 15.4 | 6.8 | 8.9 | 2.6 | 4.1 | 6.7 | 6.77 |
| ILM | 1.5 | 1.9 | 3.5 | 7.1 | 18.5 | 16.1 | 6.8 | 9.8 | 3.0 | 5.1 | 8.1 | 7.40 |
| SAE | 21.4 | 19.0 | 25.2 | 25.7 | 43.5 | 39.8 | 25.7 | 29.6 | 20.0 | 35.1 | 49.6 | 30.42 |
| ADer | 2.9 | 3.6 | 6.9 | 10.4 | 27.5 | 18.1 | 10.1 | 12.1 | 5.6 | 6.0 | 9.7 | 10.26 |
| ReColorAdv | 5.1 | 7.0 | 10.0 | 20.0 | 24.3 | 20.0 | 11.1 | 15.5 | 7.4 | 11.6 | 18.4 | 13.67 |
| cAdv | 12.2 | 14.0 | 17.7 | 11.1 | 33.9 | 32.9 | 19.9 | 23.2 | 14.6 | 16.2 | 25.3 | 20.09 |
| tAdv | 10.9 | 12.4 | 14.4 | 17.8 | 29.6 | 21.2 | 17.7 | 19.0 | 12.5 | 16.4 | 24.1 | 17.94 |
| ACE | 4.9 | 5.9 | 11.1 | 12.6 | 28.1 | 24.9 | 12.4 | 15.4 | 7.6 | 11.6 | 21.0 | 14.14 |
| ColorFool | 9.1 | 9.6 | 15.3 | 18.0 | 37.9 | 33.8 | 17.8 | 21.3 | 10.5 | 20.3 | 35.0 | 20.78 |
| NCF | 22.8 | 21.1 | 25.8 | 26.8 | 43.9 | 39.6 | 27.4 | 31.9 | 21.8 | 34.4 | 47.5 | 31.18 |
| ACA (Ours) | **52.2** | **53.6** | **53.9** | **59.7** | **63.4** | **63.7** | **59.8** | **62.2** | **53.6** | **55.6** | **60.8** | **58.05** |

setting. Specifically, when the surrogate model is RN-50, we surpass NCF by significant margins of **10.0%**, **16.1%**, **32.1%**, and **32.5%** in MobViT-s, ViT-B, Swin-B, and PVT-v2, respectively. There are two primary reasons for this phenomenon: **i)** our ACA utilizes a low-dimensional manifold search of natural images for adversarial examples, with the manifold itself determining the transferability of the adversarial examples, independent of the model's architecture; **ii)** the diffusion model incorporates the self-attention structure, exhibiting a certain degree of architectural similarity.

Overall, the deformation-based attack (ADer) exhibits lower attack performance in both white-box and black-box settings. Texture-based attacks (tAdv) show better white-box attack performance, but are less transferable than existing color-based attacks (NCF and SAE). Our ACA leverages the low-dimensional manifold of natural images to adaptively combine image attributes and generate unrestricted adversarial examples, resulting in a significant outperformance of state-of-the-art methods by **13.3%~50.4%** on average. These results convincingly demonstrate the effectiveness of our method in fooling normally trained models.

## 4.3 Attacks on Adversarial Defense Methods

The situation in that adversarial defense methods can effectively protect against current adversarial attacks exhibits the Dunning-Kruger effect [28]. Actually, such defense methods demonstrate efficacy in defending against adversarial examples within the $l_p$ norm, yet their robustness falters in the face of novel and unseen attacks [25]. Therefore, we investigate whether unrestricted attacks can break through existing defenses. Here, we choose input pre-process defenses (HGD [33], R&P [57], NIPS-r3[3], JPEG [17], Bit-Red [59], and DiffPure [39]) and adversarial training models (Inc-v3$_{ens3}$, Inc-v3$_{ens4}$, and Inc-v2$_{ens}$ [53]). Considering that some unrestricted attacks are carried out from the perspective of shape and texture, we also choose shape-texture debaised models (ResNet50-Debaised (Res-De) [32] and Shape-ResNet (Shape-Res) [14]).

The results of black-box transferability in adversarial defense methods are demonstrated in Table 2. the surrogate model is ViT-B, and the target model is Inc-v3$_{ens3}$ for input pre-process defenses. Our method displays persistent superiority over other advanced attacks by a significant margin. Our ACA surpasses NCF, SAE, ColorFool by **27.13%**, **27.63%**, and **37.27%** on average ASR. In robust models, based on $l_p$ adversarial training and shape-texture debiased models are not particularly effective and can still be easily broken by unrestricted adversarial examples. Our approach can adaptively generate various combinations of adversarial examples based on the manifold, thus exhibiting high transferability to different defense methods. Additionally, Bit-Red and Diffpure reduce the ground-truth class's confidence and increase the adversarial examples' transferability. These findings further reveal the incompleteness and vulnerability of existing adversarial defense methods.

## 4.4 Visualization

**Quantitative Comparison.** Following [47] and [60], we quantitatively assess the image quality using the non-reference perceptual image quality measure. Therefore, we choose NIMA [51], HyperIQA [49], MUSIQ [27], and TReS [15]. NIMA-AVA and MUSIQ-AVA are trained on AVA [38], and MUSIQ-KonIQ is trained on KonIQ-10K [21], following PyIQA [3]. As illustrated in Table 3, our ILM maintains the same image quality as clean images, and ACA achieves the best results in

---

[3]https://github.com/anlthms/nips-2017/tree/master/mmd

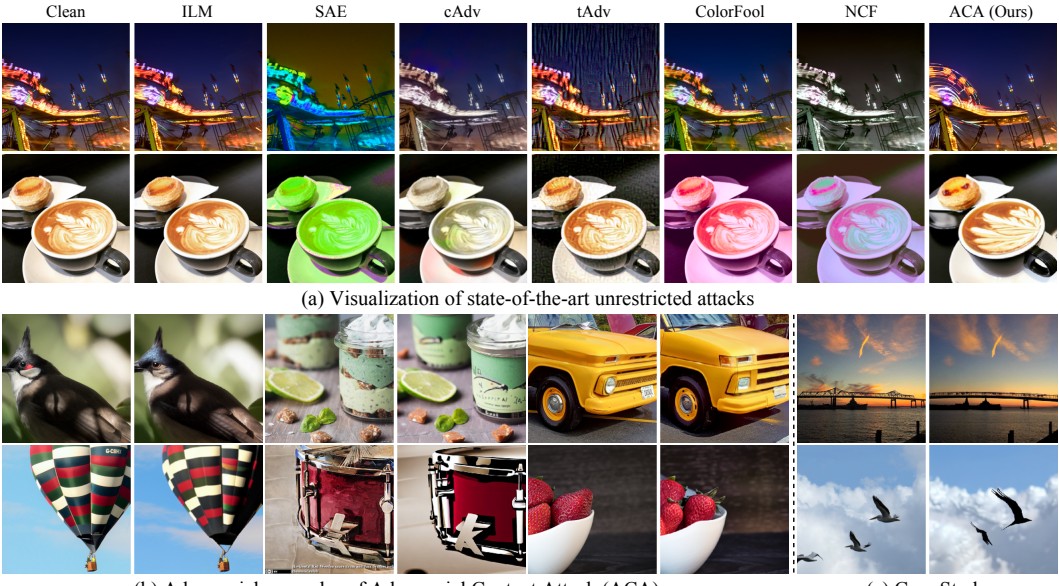

| Clean | ILM | SAE | cAdv | tAdv | ColorFool | NCF | ACA (Ours) |

(a) Visualization of state-of-the-art unrestricted attacks

(b) Adversarial examples of Adversarial Content Attack (ACA)      (c) Case Study

Figure 3: (a) Compared with other attacks, ACA generates the most natural adversarial examples; (b) ACA can generate images with various adversarial content, which can combine shape, texture, and color changes; (c) In some cases, ACA may slightly modify the semantic subject.

image quality assessment. ColorFool obtains equal or higher image quality than the clean images because it requires manually selecting several human-sensitive semantic classes and adds uncontrolled perturbations on human-insensitive semantic classes. In other words, ColorFool is bound by human intuition, so do not deteriorate the perceived image quality (our results are similar to [47]). ACA even surpasses ColorFool, mainly because: **i)** Our adversarial examples are generated based on the low-dimensional manifold of natural images, which can adaptively combine the adversarial content and ensure photorealism; **ii)** Stable Diffusion itself is an extremely powerful generation model, which produces images with very high image quality; **iii)** These no-reference image metrics are often trained on aesthetic datasets, such as AVA [38] or KonIQ-10K [21]. Some of the images in these datasets are post-processed (such as Photoshop), which is more in line with human aesthetics. Because ACA adaptively generates adversarial examples on a low-dimensional manifold, this kind of minor image editing is similar to post-processing, which is more in line with human aesthetic perception and better image quality.

**Qualitative Comparison.** We visualize unrestricted attacks of Top-5 black-box transferability, including SAE, cAdv, tAdv, ColorFool, and NCF. In Figure 3(a), we visualize adversarial examples generated by different unrestricted attacks. In night scenes and food, color and texture changes are easily perceptible, while our method still keeps image photorealism. Next, we give more adversarial examples generated by ACA. It is clearly observed that our method can adaptively combine content to generate adversarial examples, as shown in Figure 3(b). For example, the hot air balloon in the lower left corner modifies both the color of the sky and the texture of the hot air balloon. The straw-

Table 3: Image quality assessment.

| Attack | NIMA-AVA↑ | HyperIQA↑ | MUSIQ-AVA↑ | MUSIQ-KonIQ↑ | TReS↑ |
|---|---|---|---|---|---|
| Clean | 5.15 | 0.667 | 4.07 | 52.66 | 82.01 |
| ILM | 5.15 | 0.672 | 4.08 | 52.55 | 81.80 |
| SAE | 5.05 | 0.597 | 3.79 | 47.24 | 71.88 |
| ADer | 4.89 | 0.608 | 3.89 | 47.39 | 72.10 |
| ReColorAdv | 5.07 | 0.668 | 3.97 | 51.08 | 80.32 |
| cAdv | 4.97 | 0.623 | 3.87 | 48.32 | 75.12 |
| tAdv | 4.83 | 0.525 | 3.78 | 44.71 | 67.07 |
| ACE | 5.12 | 0.648 | 3.96 | 50.49 | 77.25 |
| ColorFool | 5.24 | 0.662 | 4.05 | 52.27 | 78.54 |
| NCF | 4.96 | 0.634 | 3.87 | 50.33 | 74.10 |
| ACA (Ours) | **5.54** | **0.691** | **4.37** | **56.08** | **85.11** |

berry in the lower right corner has some changes in shape and color while keeping the semantics unchanged. However, in some cases, the body of semantics changes, as shown in Figure 3(c). It may be because the prompts generated by BLIP v2 cannot describe the content of the image well.

## 4.5 Time Analysis

In this section, we illustrate the attack speed of various unrestricted attacks. We choose MN-v2 [46] as the surrogate model and evaluate the inference time on an NVIDIA Tesla A100. Table 4 shows the

Table 4: Attack speed of unrestricted attacks. We choose MN-v2 as the surrogate model and evaluate the inference time on an NVIDIA Tesla A100.

| Attack | SAE | ADer | ReColorAdv | cAdv | tAdv | ACE | ColorFool | NCF | ACA (Ours) |
|---|---|---|---|---|---|---|---|---|---|
| Time (sec) | 8.80 | 0.41 | 3.86 | 18.67 | 4.88 | 6.64 | 12.18 | 10.45 | 60.0+65.33=125.33 |

average time (in seconds) required to generate an adversarial example per image. ACA does have a significant time cost compared to other attacks. Further, we analyze the time cost and find that Image Latent Mapping (ILM) and Adversarial Latent Optimization (ALO) each accounted for 50% of the time cost. However, most of the time cost of ILM and ALO lies in the sampling process of the diffusion model. In this paper, our main contribution is to propose a new unrestricted attack paradigm. Therefore, we focus on the improvement of the attack framework, rather than the optimization of time cost. Since the time cost is mainly focused on the sampling of the diffusion model, we have noticed that many recent works have accelerated or distilled the diffusion model, which can greatly reduce the time of the sampling process. For example, [36] can reduce the total number of sampling steps by at least 20 times. If these acceleration technologies are applied to our ACA, ACA can theoretically achieve an attack speed of close to 6 seconds and we think this is a valuable optimization direction.

### 4.6 Ablation Study

The ablation studies of momentum (MO) and differentiable boundary processing (DBP) are shown in Figure 4 and the surrogate model is ViT-B [11]. *Origin* stands for ACA without MO and DBP. *MO* is Origin with momentum, and it can be observed that the adversarial transferability is significantly improved after the introduction of momentum. *MO+DBP* is Origin with momentum and DBP. Since DBP further optimizes the effectiveness of the adversarial examples and constrains the image within the range of values, it can still

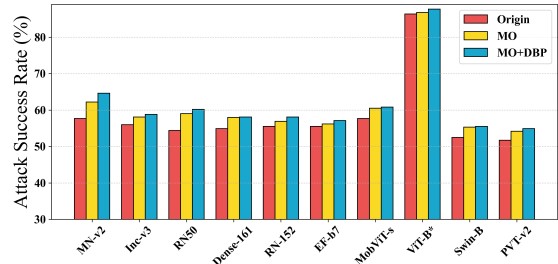

Figure 4: Ablation studies of momentum (MO) and differentiable boundary processing (DBP).

improve the adversarial transferability. Although the above strategies are not the main contribution of this paper, the above experiments illustrate that they can boost adversarial transferability.

## 5 Conclusions

In this paper, we propose a novel unrestricted attack framework called Content-based Unrestricted Adversarial Attack. We map the image onto a low-dimensional manifold of natural images. When images are moved along the adversarial gradient on the manifold, unrestricted adversarial examples with diverse adversarial content and photorealistic can be adaptively generated. Based on the diffusion model, we implement Adversarial Content Attack. Experiments show that the existing defense methods are not very effective against unrestricted content-based attacks. We propose a new form of unrestricted attack, hoping to draw attention to the threat posed by unrestricted attacks.

**Limitations.** Due to the inherent limitations of diffusion models, a considerable number of sampling steps are required in the inference process, resulting in a relatively longer runtime, taking $\sim 2.5$ minutes using a single A100 GPU. Additionally, the current adaptive generation of unrestricted adversarial examples does not allow for fine-grained image editing. These issues are expected to be resolved in the future with further advancements in diffusion models.

**Negative Social Impacts.** Adversarial examples from our method exhibit a high photorealism and strong black-box transferability, which raises concerns about the robustness of DNNs, as malicious adversaries may exploit this technique to mount attacks against security-sensitive applications.

**Acknowledgements.** This work was supported by National Natural Science Foundation of China (No.62072112), Scientific and Technological innovation action plan of Shanghai Science and Technology Committee (No.22511102202).

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
