# OpenReview forum: "Content-based Unrestricted Adversarial Attack"
_NeurIPS.cc/2023/Conference — NeurIPS 2023 poster_

### Official Review · Reviewer_QZtm · 2023-07-04

**Soundness:** 3 good
**Presentation:** 3 good
**Contribution:** 3 good
**Rating:** 6
**Confidence:** 5

**Summary:**

This paper proposes a novel unrestricted attack framework called Content-based Unrestricted Adversarial Attack (ACA). The author argues that current unrestricted attacks have limitations in terms of maintaining human visual imperceptibility, generating natural adversarial examples, and achieving high attack performance. This paper is well-written and easy to follow.

**Strengths:**

1) The paper introduces a novel attack framework that addresses the limitations of current unrestricted attacks. The use of a low-dimensional manifold and optimization along its adversarial direction allows for the generation of diverse and natural adversarial examples.
2) The paper provides a clear motivation and problem definition, outlining the challenges and goals for unrestricted adversarial attacks.
3) The paper includes extensive experimentation and visualization to validate the effectiveness of ACA. The results show significant improvements over state-of-the-art attacks in terms of adversarial transferability.

**Weaknesses:**

The paper does not include sufficient evaluation and ablation studies for the proposed method. Since ACA used skip gradient, I think I should compare Skip connections matter [1] in the experiment.

The paper could benefit from a more thorough review of relevant literature. While the authors mention existing unrestricted attacks, there is limited discussion on related work and the novelty  of ACA compared to previous approaches.



[1] Skip connections matter: On the transferability of adversarial examples generated with resnets, ICLR 2019

**Questions:**

/Na

**Limitations:**

/Na

---

> ### Author Rebuttal · Authors · 2023-08-09
>
> Thanks for your valuable comments. What excites us is not the high rating you have given us, but the questions you have proposed. This assures us that you have a deep knowledge of the field and have recognized the significance of our work to the community.  **Therefore, we hope you can champion our work in the discussion.** We are convinced that our work is worthy of being presented to a wider audience of researchers and can assist ML systems in mitigating the threat of unrestricted attacks in practice. We address your concerns as follows:
>
> **[Q1: Insufficient evaluation and ablation studies.]** Based on your valuable suggestions, we add an experimental comparison with SGM in Q2. For more evaluation, we add more attack and defense experiments (please refer to **Reviewer frod #Q1**). In the supplementary material, we provide ablation studies with momentum, differentiable boundary processing, and $\beta$. Further, we also supplement the ablation studies of momentum factor $\mu$ and perturbation value $\kappa$ (please refer to **Author Rebuttal #Q3**).
>
> **[Q2: Comparison on Skip connections matter (SGM).]** Thank you for pointing out that this is actually a duplication of terms, that is, our skip gradient (SG) is similar in name to the Skip connections matter (SGM) you mentioned, but they are technically different.
> - In terms of purpose, our SG is to solve the problem of memory overflow caused by gradient backpropagation during the sampling process of the diffusion model, but SGM is using more gradients from the skip connections rather than the residual modules to enhance transferability.
> - In terms of implementation, SG uses the derivative in the denoising process to approximate the gradient, while SGM uses a single decay factor to enhance the gradient on skip connections.
> - In terms of scope of application, SG is suitable for diffusion models, while SGM is suitable for neural networks with skip connections.
>
> Overall, both SG and SGM directly or indirectly improve adversarial transferability. We will add a corresponding discussion to illustrate the above issues in the final version. Furthermore, we supplement related experiments following your suggestion. Because SGM is only effective for gradient optimization-based methods and is a plug-and-play module, we choose ADer, ReColorAdv, cAdv, ACE and ACA for experiments (ResNet50 is the surrogate model). Experiments show that SGM has almost no effect on ADer and cAdv, but can significantly improve transferability on other methods (ACA improves Avg. ASR by 5.87%).
>
>  Attack          | MN-v2 | Inc-v3 | RN-50 | Dense-161 | RN-152 | EF-b7 | MobViT-s | ViT-B | Swin-B | PVT-v2 | Avg. ASR (%)
> :---------------:|:-----:|:------:|:-----:|:---------:|:------:|:-----:|:--------:|:-----:|:------:|:------:|:-------------:
>  ADer            | 15.5  | 7.7    | 55.7* | 8.4       | 7.8    | 11.4  | 12.3     | 9.2   | 4.6    | 4.9    | 9.09
>  ADer+SGM        | 14.8  | 7.3    | 10.5* | 7.7       | 7.4    | 11.1  | 11.8     | 9.5   | 4.5    | 4.7    | 8.76
>  ReColorAdv      | 40.6  | 17.7   | 96.4* | 28.3      | 33.3   | 19.2  | 29.3     | 18.8  | 12.9   | 13.4   | 23.72
>  ReColorAdv+SGM  | 58.3  | 26.7   | 98.6* | 48.4      | 53.0   | 25.8  | 43.5     | 23.8  | 21.1   | 20.9   | 35.72
>  cAdv            | 44.2  | 25.3   | 97.2* | 36.8      | 37.0   | 34.9  | 40.1     | 30.6  | 19.3   | 20.2   | 32.04
>  cAdv+SGM        | 43.4  | 25.5   | 57.8* | 32.2      | 30.3   | 31.9  | 39.0     | 30.9  | 19.3   | 19.5   | 30.22
>  ACE             | 32.8  | 9.4    | 99.1* | 16.1      | 15.2   | 12.7  | 20.5     | 13.1  | 6.1    | 5.3    | 14.58
>  ACE+SGM         | 61.4  | 20.9   | 99.2* | 33.8      | 37.8   | 20.8  | 36.3     | 26.0  | 11.7   | 10.0   | 28.74
>  ACA (Ours)      | 69.3  | 61.6   | 88.3* | 61.9      | 61.7   | 60.3  | 62.6     | 52.9  | 51.9   | 53.2   | 59.49
>  ACA (Ours) +SGM | 76.7  | 67.0   | 91.7* | 68.8      | 69.0   | 59.9  | 68.5     | 58.5  | 58.2   | 61.6   | 65.36
>
>
> **[Q3: Thorough review of relevant literature.]** Thanks for pointing out related work and novelty issues discussed in the current version. **For related work**, we briefly summarize the shortcomings of current shape-, texture-, and color-based unrestricted attacks in Lines 35-45. Then we further analyze the shortcomings of ColorFool and Natural Color Fool in Lines 86-96. **For the novelty of ACA**, we summarize the innovations in the attack form in Lines 97-100: ACA can adaptively combine multiple contents (shape, texture and color), guarantee the photorealism of the image, and exhibit potential attack performance. In Section 3 , we demonstrate the technical innovations of ACA, including Image Latent Mapping and Adversarial Latent Optimization. Following your comments, we will add detailed descriptions of each unrestricted attack, systematically summarize the shortcomings, and discuss the differences with ACA in more detail to highlight the novelty of ACA. Thank you again for your help in improving the quality of the paper.

---

> > ### Comment · Reviewer_QZtm · 2023-08-20
> > **Response to authors**
> >
> > The author answered my questions and I will improve my score.

---

### Official Review · Reviewer_frod · 2023-07-04

**Soundness:** 3 good
**Presentation:** 3 good
**Contribution:** 3 good
**Rating:** 5
**Confidence:** 4

**Summary:**

The paper introduces a novel attack framework called Content-based Unrestricted Adversarial Attack, which aims to generate diverse and natural adversarial examples with high transferability. The authors argue that existing methods, such as lp norm-based attacks, have limitations in terms of perceptual similarity, naturalness, and robustness. To address these issues, they propose mapping images onto a low-dimensional manifold represented by a generative model trained on natural images. This manifold ensures both photorealism and content diversity. By optimizing the adversarial objective on this latent space, they generate unrestricted adversarial examples. The proposed method, called Adversarial Content Attack (ACA), utilizes Image Latent Mapping (ILM) and Adversarial Latent Optimization (ALO) techniques to optimize the latent in a diffusion model. The effectiveness of ACA is validated through experiments and visualization, demonstrating significant improvements of 13.3~50.4% in terms of adversarial transferability compared to state-of-the-art attacks.

Overall, the main contributions of the paper are the introduction of the Content-based Unrestricted Adversarial Attack framework, the development of the Adversarial Content Attack method, and the experiments demonstrating improvements in generating diverse and transferable adversarial examples.

**Strengths:**

The authors effectively communicated the motivation, problem statement, and methodology of the proposed framework. By addressing the limitations (imperceptibility/photorealism/effectiveness) of existing methods, they proposed a novel attack framework that leverages a low-dimensional manifold represented by a generative model. By combining image mapping onto a latent space, optimizing adversarial objectives, and utilizing a diffusion model, the authors introduce a novel approach to generating diverse and natural adversarial examples. This paper might be the first to explore unrestricted adversarial examples through such a framework.

This paper offers a thorough explanation of the proposed attack framework, detailing the underlying techniques of ILM and ALO. The authors further support their claims through experimentation and visualization, providing evidence of the effectiveness of their approach and demonstrating improvements in adversarial transferability compared to state-of-the-art attacks. The improvements in adversarial transferability also shed light on the potential impact of this method in uncovering vulnerabilities in security-sensitive applications and advancing our understanding of robustness in DNNs.

Overall, this paper's strengths encompass originality in proposing a novel attack framework, quality in terms of methodology and experimental evaluation, clarity in explaining the concepts and techniques, and significance in addressing limitations and raising awareness of unrestricted but realistic adversarial examples.


**Weaknesses:**

While the paper has several strengths, there are some weaknesses that could be addressed to further improve the work:

**Comparison with State-of-the-Art Attacks:** While the paper mentions that the proposed method achieves significant improvements in terms of adversarial transferability compared to state-of-the-art attacks, a more comprehensive comparison would strengthen the evaluation. It would be valuable to include a thorough analysis and comparison with a wider range of existing unrestricted attack methods, such as [1]. In particular, Laidlaw et al. proposed an efficient way to generate imperceptible adversarial examples. The reviewer also suggests evaluating the proposed attack on other adversarially trained models, for example, the defense method that could be generalized to unforeseen perturbations [1], or having used synthetic data during adversarial training [2].

**Defense Method:** This paper does not discuss how to defend the proposed attacks but primarily on the efficacy of the proposed method. It would be great if the authors provide potential solutions or mitigation strategies for the threats.

**Generalization to Different Datasets:** The authors only evaluate their method on a subset of the ImageNet validation set and do not say how the results generalize to other datasets. It would be beneficial to investigate the generalization of the proposed approach to different datasets.

**Unclear Claim:** The authors mentioned the Dunning-Kruger effect to emphasize that current defense methods against lp norm adversarial examples overestimate their abilities. However, this work does not provide further details and arguments to support this. Although similar arguments have also been proposed in [3], in which the authors argue that lp-based robustness evaluation might be biased, the reviewer thinks that using the Dunning-Kruger effect here is not rigorous. The reviewer suggests that the authors rethink this argument, and even consider removing it if it is not an important contribution of the paper.

**Typos:** Although this paper is well-written and easy to follow, the reviewer found some typos and grammar errors. For example, in line 161, *follw*. It would be great if the authors have proofread the paper before submitting it.

    [1] Laidlaw et al. Perceptual adversarial robustness: Defense against unseen threat models. (ICLR 2021)
    [2] Croce et al. Robustbench: a standardized adversarial robustness benchmark. (NeurIPS 2021)
    [3] Hsiung et al. CARBEN: Composite Adversarial Robustness Benchmark. (IJCAI 2022)

**Questions:**

Please refer to the weakness part. In brief, the reviewer would like to understand more about the following points:

- Could the authors provide more attack and defense baselines as mentioned in the weakness?
- Could the authors provide some experiments on other datasets?
- Please address the mentioned unclear claim.
- Why the authors did not provide the code for review but answered the Reproducibility as "yes"?

---

> ### Author Rebuttal · Authors · 2023-08-09
>
> Thank you for your time and effort in our work. We supplement the experiments with the attack and defense, and release the code. Hope you can further support our work.
>
> **[Q1: Comparison with state-of-the-art attacks.]** Thanks for your valuable suggestions, we supplement the attack experiments of Perceptual Projected Gradient Descent (PPGD) and Lagrangian Perceptual Attack (LPA) in [1]. PPGD exhibits poor attack performance, while LPA has better resistance to migration. However, compared with the state-of-the-art NCF and ACA, there is still a large gap (ACA exceeds 34.47% on Avg. ASR).
>
>  Attack     | MN-v2 | Inc-v3 | RN-50 | Dense-161 | RN-152 | EF-b7 | MobViT-s | ViT-B | Swin-B | PVT-v2 | Avg. ASR (%)
> :----------:|:-----:|:------:|:-----:|:---------:|:------:|:-----:|:--------:|:-----:|:------:|:------:|:-------------:
>  PPGD       | 23.1  | 12.3   | 99.7* | 16.6      | 18     | 13.3  | 14.9     | 10.6  | 6.3    | 6.9    | 13.56
>  LPA        | 37.6  | 24     | **100***  | 34.4      | 38     | 22    | 29.2     | 13.5  | 12.2   | 14.3   | 25.02
>  NCF        | **71.2**  | 33.6   | 91.4* | 48.5      | 60.5   | 32.4  | 52.6     | 36.8  | 19.8   | 21.7   | 41.90
>  ACA (Ours) | 69.3  | **61.6**   | 88.3* | **61.9**      | **61.7**   | **60.3**  | **62.6**    | **52.9**  | **51.9**   | **53.2**   | **59.49**
>
> Furthermore, we complement the defense experiments on adversarially trained models. Since the pre-trained robust model weights are not released in [1], we cannot reproduce the adversarial training model on ImageNet quickly due to time and computing power. Therefore, we choose ViT -B-CvSt and ConvNext-L-CvSt in state-of-the-art on the leaderboard in [2] (the model comes from [A3] and the input size is 224). Our ACA still outperforms other unrestricted attacks by a significant margin.
>
>  Attack         | Clean | ILM  | SAE   | ADer | ReColorAdv | cAdv  | tAdv  | ACE   | ColorFool | NCF  | ACA (Ours)
> :--------------:|:-----:|:----:|:-----:|:----:|:----------:|:-----:|:-----:|:-----:|:---------:|:----:|:----------:
>  ViT-B-CvSt      | 8.4   | 8.7  | 38.9  | 11.2 | 10.5       | 20.6  | 11.3  | 16.9  | 31.2      | 35.8 | **51.1**
>  ConvNext-L-CvSt  | 7.4   | 7.9  | 34.5  | 11.0   | 9.5        | 17.3  | 10.7  | 15.4  | 26.9      | 33.0   | **49.7**
>
> Thanks again for your suggestion, we will add this part of the experiment to the final version.
>
> [A3] Revisiting Adversarial Training for ImageNet: Architectures, Training and Generalization across Threat Models, arXiv preprint arXiv:2303.01870.
>
> **[Q2: Defense method.]** We believe that improving the adversarial robustness of the model itself is a potential defense strategy. First, recent work on LLM [A4, A5] shows that larger models are more robust. Therefore, the construction of the foundation model may be a potential direction. Secondly, we can consider designing specific training strategies, such as the one you mentioned [1]. It may be an idea to add unrestricted adversarial examples to adversarial training. In the end, we thought that maybe a better way of visual encoding might also be a solution. We will incorporate these ideas into the final version.
>
> [A4] Nicholas Carlini et. al., Are aligned neural networks adversarially aligned?
> [A5] Jindong Wang et. al., On the Robustness of ChatGPT: An Adversarial and Out-of-distribution Perspective
>
> **[Q3: Generalization to different datasets.]** We choose ImageNet mainly to set up alignment with other methods, which is conducive to a fair comparison. In terms of models, ImageNet has the most image classification models, which is convenient for the experiment of transfer attack; in terms of difficulty, ImageNet has 1000 categories, rich in content and semantics, which are more in line with images in real scenes, so experiments on this dataset can make the results generalizable. In addition, in the field of transfer attacks, ImageNet-compatible Dataset is widely used. We admit that our method will have limitations on small-sized images, such as CIFAR because it requires a larger-sized input, but the results on ImageNet have been able to illustrate the effectiveness and generalization of ACA. If time permits, we will add experiments on more datasets in the future.
>
> **[Q4: Unclear claim.]** A recent paper [A6] describes the Dunning-Kruger effect as a cognitive bias in that humans tend to overestimate their abilities. We argue that the adversarial robustness of current adversarial defenses is currently overestimated. In fact, the current adversarial defense can only better defend against the $l_p$ adversarial example, but cannot effectively defend against unrestricted adversarial examples. We detail the experimental results of attacking defenses in Section 4.3. ACA achieves a transfer attack success rate of more than 50% in all defense methods. We think this greatly illustrates the insufficiency of current adversarial defenses in the face of unrestricted adversarial examples, so we interpret this overestimation as a Dunning-Kruger effect. Regarding this statement, we are happy to continue to communicate with you in the discussion session.
>
> [A6] Reliability in Semantic Segmentation: Are We on the Right Track? CVPR 2023
>
> **[Q5: Typos.]** Thank you for your detailed review of our work, which is of great help to us in improving the quality of the paper. We will carefully check all typos and revise them in the final version.
>
> **[Q6: Reproducibility as "yes".]** Following the NeurIPS Paper Checklist Guidelines, our paper provides enough information to reproduce the experiment, so we choose "yes". Also during the rebuttal, we submitted the anonymous link containing the code to AC. In order to promote the research on unrestricted attacks, we promise to release the code after accepting this work.

---

> > ### Author Response · Authors · 2023-08-15
> > **Happy to Discuss with You**
> >
> > Dear Reviewer frod:
> >
> > As the discussion period is closing, we sincerely look forward to your feedback. We deeply appreciate your valuable time and efforts spent reviewing this paper and helping us improve it.
> >
> > It would be very much appreciated if you could once again help review our responses and let us know if these address or partially address your concerns and if our explanations are heading in the right direction.
> >
> > Please also let us know if there are further questions or comments about this paper. We strive to improve the paper consistently, and it is our pleasure to have your feedback!
> >
> > Best regards,
> >
> > Authors

---

> > ### Comment · Reviewer_frod · 2023-08-15
> >
> > I appreciate the authors' efforts in addressing my questions. While a significant portion of my concerns has been satisfactorily addressed, I find that the conclusion drawn regarding the application of the Dunning-Kruger effect in the context of current defense methods remains somewhat unclear to me. It's important to note that the Dunning-Kruger effect is commonly employed to elucidate cognitive biases exhibited by humans. This is also not surprising that the proposed method could be cracked by unseen attacks.
> >
> > Given that the Dunning-Kruger effect has been previously elucidated in [A6], the reviewer acknowledges that this particular concern is minor. I have raised my score. However, the reviewers suggest the authors discuss these previous works in the revision to make the claims more rigorous and substantiated.

---

### Official Review · Reviewer_Rhy1 · 2023-07-06

**Soundness:** 2 fair
**Presentation:** 2 fair
**Contribution:** 2 fair
**Rating:** 6
**Confidence:** 3

**Summary:**

In this paper, authors propose an unrestricted untargeted attack based on optimising the latent space of stable diffusion model. The generated adversarial samples are empirically shown to be more transferable than the existing semantic attacks. Moreover, authors validate the effectiveness of adversarial samples when attacking different representative adversarial trained models and show consistent performance book.

**Strengths:**

- The method is a replacement of popular GANs in the semantic attacks with the powerful stable diffusion models, pushing the state-of-the-art by many steps emperically.
- Experiments are thorough in attacking both normal and adversarial trained models
- Compared against many baseline unrestricted attacks and proposed attacks outperforms all the methods.

**Weaknesses:**

-  The paper  lacks technical novelty as the latent space optimisation for generating adversarial attacks as been popular from many years [17].

- Since this is a unrestricted attack, the generated image quality is difficult to assess with metrics. Nonetheless, authors have computed 5 metrics to understand the image quality

- Authors did not discuss about the code release to reproduce the experiments. For this particular paper, the implementation of this proposed method is not simple as the section 3.1 and 3.2 mostly discusses about the difficulties in optimising the latent space and how they overcome it through skip gradient, momentum and boundary function.

**Questions:**

Overall, I do not have major questions as the proposed method is properly motivated and leverages the powerful generative model to aid the attack generation. However, I have few minor questions mostly around implementation and design choices:

- In section 3.1, authors propose to optimise the null text embedding $∅_t$ at every timestamp to offset the error. Is there any difference to your implementation as compared to the method of [32]. Can you please clarify your contribution in the section 3.1 ie. mapping image $z_0$ to latent space $z_t$.

- In particular, authors show the benefit of momentum factor $\mu$  and boundary function $ϱ(·)$ to improve the ASR in Appendix D.  Did you perform ablation study to set the value of  $\mu$ to 1 and can authors provide more insights on the design of boundary function in particular for inputs outside the valid range [0, 1]?

- How do you ensure the perturbation value of $k = 0.1$ in the latent space is not drastically modifying the original image.  Can you present an ablation of this key parameter vs attack performance vs image quality ?  In Table 3, there is no discussion about the non-reference metrics mentioned in the paper. Why is the method not evaluated with metrics such as LPIPS that captures perceptual similarity? Please also include FID to the metrics.

- How is the performance of the attacks in targeted attack setting? Do you require larger shift in latent space to craft example?

- Please also benchmark the baseline attacks in terms of attack speed for completeness.

- Can you constrain the perturbation to a local region in latent space to generate a kind of patch attacks in image space?

Finally, I request the authors to discuss their plans in releasing the codebase to reproduce the experiments as I believe this will be invaluable to the community.

**Limitations:**

- The inference time of the attack is much higher taking 2.5 minutes for image due to many many levels of optimisation such as for null text embedding, perturbation in latent space.

---

> ### Author Rebuttal · Authors · 2023-08-09
>
> Thanks for your valuable comments and we address your concerns as follows.
>
> **[Q1: Technical novelty of latent space optimization.]** Techniques for latent space optimization using Generative Adversarial Networks (GAN) are common, but latent optimization using diffusion models has not been widely explored.
>
> - First, the latent of GAN and diffusion model is different. The latent of GAN is usually decoupled and limited to certain attributes, while the latent of the diffusion model is aligned with the text. We choose the null text embedding for optimization, propose a corresponding attack algorithm, and illustrate through experiments that it can generate high-transferability adversarial examples.
>
> - Then, we find that the sampling process of the diffusion model is unable to construct the calculation graph of the gradient chain, because the GPU memory will overflow (Lines 192-198), so we propose the skip gradient to solve the unique problem of this diffusion model.
>
> Therefore, our work is designed for the latent of the diffusion model, which is the biggest difference between us and the previous work on latent space optimization.
>
> **[Q2: Choice of image quality assessment.]** Please refer to **Author Rebuttal #Q1**.
>
> **[Q3: Comparison on [32] and contributions.]** The biggest contribution of this paper is to propose a novel unrestricted attack framework called Content-based Unrestricted Adversarial Attack. Under this framework, we first employ Image Latent Mapping (ILM) to map images onto the latent space and utilize Adversarial Latent Optimization (ALO) to optimize the latent. We emphasize that our contribution is to introduce the ILM module to realize ACA so that images can be mapped to latent space to complete subsequent attacks. The existence of this model is very necessary, and [32] is just one implementation of ILM. This implementation can be replaced, or even superseded by better methods. Compared to other strategies [9, 39, 23, 49], the current strategy [32] is simple and effective and does not require fine-tuning to obtain high-quality image reconstruction. Our framework is not limited to this implementation, and the proposed framework can still be applied if there are better implementations in the future.
>
> **[Q4: Abaltion study on momentum factor.]** Please refer to **Author Rebuttal #Q3**.
>
> **[Q5: Insight about boundary function.]** The motivation of Differentiable Boundary Processing is to ensure that the value range of adversarial examples is between [0,1]. Because when the diffusion model generates an image, the value range may not be in [0,1]. When saving adversarial examples, it is usually directly cropped to [0,1], and the part of the perturbation that is not in [0,1] is ignored, which may cause the attack to fail. To reduce the ASR reduction caused by the error during storage, we use DBP to constrain the value range of the adversarial examples to [0,1] as much as possible.
>
> **[Q6: Perturbation value.]** Please refer to **Author Rebuttal #Q3**.
>
> **[Q7: Targeted attacks.]** Please refer to **Author Rebuttal #Q2**.
>
> **[Q8: Attack speed.]** Here, we illustrate the attack speed of various unrestricted attacks. We choose MN-v2 as the surrogate model and evaluate the inference time on an NVIDIA Tesla A100. The table shows the average time (in seconds) required to generate an adversarial example per image. ACA does have a significant time cost compared to other attacks. Further, we analyze the time cost and find that Image Latent Mapping and Adversarial Latent Optimization each accounted for ~50% of the time cost. However, most of the time cost of ILM and ALO lies in the sampling process of the diffusion model. We also take the initiative to explain this issue in the limitation.
>
>  Attack      | SAE    | ADer   | ReColorAdv | cAdv    | tAdv   | ACE    | ColorFool | NCF     | ACA (Ours)
> :-----------:|:------:|:------:|:----------:|:-------:|:------:|:------:|:---------:|:-------:|:----------:
>  Time (sec)  | 8.80   | 0.41   | 3.86       | 18.67   | 4.88   | 6.64   | 12.18     | 10.45   | 60.0+65.33=125.33
>
> **In this paper, our main contribution is to propose a new unrestricted attack paradigm. Therefore, we focus on the improvement of the attack framework, rather than the optimization of time cost.** Since the time cost is mainly focused on the sampling of the diffusion model, we have noticed that many recent works have accelerated or distilled the diffusion model, which can greatly reduce the time of the sampling process. For example, [A2] can reduce the total number of sampling steps by at least 20 times. If these acceleration technologies are applied to our ACA, ACA can theoretically achieve an attack speed of close to 6 seconds. Thank you for your valuable suggestions, we think this is a valuable optimization direction.
>
> [A2] On Distillation of Guided Diffusion Models, CVPR 2023
>
> **[Q9: Local edit.]** Local editing is currently not supported in this paper. But with the rapid development of diffusion models and the emergence of more control and editing techniques, we believe that it is possible to incorporate local editing into our proposed unrestricted attack framework in the future. In addition, this issue has been displayed in the main body as a limitation.
>
> **[Q10: Release codes.]** Thank you for recognizing the value of our work. During the rebuttal, we have submitted an anonymous link with the codes to AC. To promote the research on unrestricted attacks, we promise to release the codes after accepting this work.

---

> > ### Comment · Reviewer_Rhy1 · 2023-08-21
> >
> > Thank you authors for the rebuttal and additional experiments. My concerns regarding momentum $\mu$ and perturbation value $k$ is addressed. The rebuttal experiments suggest that the attack success rate is not overly sensitive to these hyper-parameters. Moreover, I believe the boundary function is an additional contribution which can be extended to other attacks.
> >
> > On the other hand, I request the authors to incorporate the experiments about the attack speed in the revised paper and share an  user-friendly reproducible code for the benefit of community.
> >
> > Overall, I believe this paper will inspire the future research on diffusion-based unrestricted attacks and will be of interest to the audience at NeurIPS. I increase my score to WA.

---

### Official Review · Reviewer_oMF6 · 2023-07-14

**Soundness:** 3 good
**Presentation:** 3 good
**Contribution:** 2 fair
**Rating:** 6
**Confidence:** 4

**Summary:**

This paper proposes the Adversarial Content Attack based on the diffusion model. The proposed attack method first maps the image onto a low-dimensional manifold of natural images and then moves images along the adversarial gradient on the manifold to generate photorealistic adversarial examples. The authors conduct extensive experimentation and demonstrate the efficacy of the Adversarial Content Attack in both normally trained models and defense methods.

**Strengths:**

1. The paper presents a host of visualization and experiments to support the intuitions and conclusions.
2. The investigated topic is important and useful.

**Weaknesses:**

1. This work seems like an application of the diffusion model in model debugging research, finding some hard samples of deep models. However, it lacks mentions or comparisons with related work, including [1],[2], etc.

2. In Table 3, I can hardly understand why the generated adversarial image can achieve better photorealism than the real image.

3. In Table 2, Inc-v3$_{ens4}$ is used as the target model, which is a defense method. But confusingly, the attack success rate (**62.2**) surpasses the case when using normal Inc-v3 (**58.8**) as the target.

I will be pleased to raise my score if these questions can be properly answered.

[1] Xiaodan Li, Yuefeng Chen, Yao Zhu, Shuhui Wang, Rong Zhang, Hui Xue. ImageNet-E: Benchmarking Neural Network Robustness via Attribute Editing.  In Proceedings of the IEEE/CVF Conference on Computer Vision and Pattern Recognition, 2023.

[2] Maximilian Augustin, Valentyn Boreiko, Francesco Croce, Matthias Hein. Diffusion Visual Counterfactual Explanations. Annual Conference on Neural Information Processing Systems 2021, NeurIPS 2021.

**Questions:**

This paper mainly focuses on the untargeted attack. I wonder whether the proposed attack method can also be applied to the targeted attack case, that is, to specify the misclassified category.

**Limitations:**

The paper has discussed the limitations and negative social impacts of the proposed attack method.

---

> ### Author Rebuttal · Authors · 2023-08-09
>
> We thank you for valuable advice. We will follow your advice to complement the discussion of related work and the explanation of image quality. We would appreciate it very much if you can champion us.
>
> **[Q1: Related works about model debugging.]** Thanks for your help in improving the quality of our paper. ImageNet-E [1] is released after the NIPS submission is over, so we do not discuss it in time. We will add the following discussion to the final version:
>
> - The image editing of ImageNe-E [1] focuses on the single element that explicitly controls the image. It evaluates the robustness of background, size, pose, and direction, and does not emphasize the resulting adversarial robustness. However, our method can adaptively and implicitly generate adversarial examples with shapes, colors, and textures, with a variety of adversarial content, emphasizing the adversarial robustness. Further, it is found through experiments that it also has strong adversarial transferability. The focus is different on control methods, elements and robustness.
>
> - DVCE [2] is similar to ours in terms of generation effect, but in terms of technical route, optimization strategy and parameter update are different. The proposed Cone Projection is only suitable for robust classifiers, while our method does not have any requirements on the model. In addition, model debugging focuses on the bias of the classifier, such as some spurious features. While ACA pays more attention to the adversarial vulnerability of the model, expecting to find small modified error examples.
>
> **[Q2: Explanation on image quality.]** Thanks to your serious review, we have also noticed that the adversarial examples generated by ACA are more photorealism than real images, and explained the possible reasons in Lines 296-304:
>
> - Our adversarial examples are generated based on the low-dimensional manifold of natural images, which can adaptively combine the adversarial content and ensure photorealism;
> - Stable Diffusion itself is an extremely powerful generation model, which produces images with very high image quality.
>
> It should be noted that this is not the first time a similar situation has appeared in this work. It also appeared in ColorFool (CVPR2020), where the unrestricted adversarial examples have better quality than the original image. In addition to the above 2 points, we further analyze the possible reasons. These no-reference image metrics are often trained on aesthetic datasets, such as ACA or KonIQ-10K. Some of the images in these datasets are post-processed (such as Photoshop), which is more in line with human aesthetics. Because ACA adaptively generates adversarial examples on a low-dimensional manifold, this kind of minor image editing is similar to post-processing, which is more in line with human aesthetic perception and better image quality. We will update the explanation of this part in the final version.
>
> **[Q3: ASR on defense models.]** Inc-v3$\_{ens4}$ is a robust model based on adversarial training and adversarial training generally reduces the clean accuracy of the model (compared to the normal training model). Furthermore, Inc-v3$\_{ens4}$ is based on $l_p$ perturbation for adversarial training, which does not have a good defense effect on unrestricted adversarial examples [55]. This experiment also verifies that the existing adversarial examples provide false adversarial robustness.
>
> **[Q4: Targeted attacks.]** Please refer to **Author Rebuttal #Q2**.

---

> > ### Comment · Reviewer_oMF6 · 2023-08-15
> >
> > The authors have addressed my concerns and I appreciate the efforts the authors made to refine the paper. I have raised my score. Though I recommend this paper to be accepted, I am also willing to hear about the other reviewers' further opinions and discussion.

---

### Author Rebuttal · Authors · 2023-08-09

We sincerely appreciate all reviewers' valuable feedback and the efforts of the program chair and area chair. So, we are committed to addressing the issues you raised and improving our manuscript accordingly. Next, we provide point-to-point responses to each reviewer and address all details.

**[Q1: Choice of image quality assessment.]** The quality assessment of generated images has always been a concern of the academic community, and numerous endeavors have been undertaken to tackle the issue. We acknowledge that objective metrics alone cannot perfectly reflect the image quality of unrestricted adversarial examples, so we provide qualitative and quantitative analysis in Section 4.4 to analyze image quality. The reason why we did not choose common perceptual image quality metrics such as LPIPS or FID is that the backbone (AlexNet/VGG/InceptionV3) they use is pre-trained on ImageNet and vulnerable to adversarial attacks, which may lead to biased image quality estimation. So we don't provide relevant values. Further, the unrestricted attacks recently published at the top tier conference like ColorFool (CVPR2020) and NCF (NIPS2022), both choose NIMA, a non-reference perceptual image quality measure, as the evaluation standard. As this metric has been rigorously peer-reviewed and verified to be feasible, we supplemented the NIMA metric with four additional metrics to achieve a more comprehensive evaluation of the visual quality. The proposed evaluation scheme may not be the optimal solution, but we believe that it is a relatively appropriate strategy at the moment.

**[Q2: Targeted Attack.]** Yes, our ACA can achieve targeted attacks by modifying the loss function, just like PGD. Considering that previous unrestricted attacks pay more attention to untargeted adversarial transferability, our experiments are also mainly aligned with previous work. Further, we implement this targeted attack, integrate this attack into codes and submit it to AC. Therefore, we hope that reviewers will support our work to advance the field's attention to the threat of unrestricted attacks.

**[Q3: More abaltion study.]** Following **Reviewer Rhy1**'s suggestion, we supplement the ablation study of momentum factor $\mu$ and perturbation value $\kappa$ (MN-v2 as the surrogate model).

- **Momentum factor $\mu$**: As $\mu$ becomes greater, the black-box average ASR will rise first. It peaks at $\mu = 1$, greater $\mu$ leads to a slight drop in performance. It is worth noting that **even without momentum** ($\mu=0$), our ACA also outperforms the next highest attack (NCF) by 12.04%.

 $\mu$  | MN-v2 | Inc-v3 | RN-50 | Dense-161 | RN-152 | EF-b7 | MobViT-s | ViT-B | Swin-B | PVT-v2 | Avg. ASR (%)
:---:|:-----:|:------:|:-----:|:---------:|:------:|:-----:|:--------:|:-----:|:------:|:------:|:-------------:
 0   | 91.8*  | 53.1   | 58.9  | 55.2      | 54.7   | 53.4  | 56.2     | 47.8  | 45.1   | 46.5   | 52.32
 0.2 | 93.1* | 55.1   | 60.4  | 54.0        | 53.6   | 50.5  | 55.1     | 45.9  | 43.6   | 46.8   | 51.67
 0.4 | 93.5*  | 53.7   | 60.3  | 53.6      | 53.9   | 52.2  | 57.5     | 47.9  | 46.1   | 46.4   | 52.40
 0.6 | 92.9*  | 54.5   | 59    | 56.2      | 56.1   | 52.7  | 58.9     | 48.6  | 47.0     | 47.7   | 53.41
 0.8 | 92.7*  | 55.6   | 59.4  | 56.5      | 56.3   | 52.7  | 57.6     | 50.5  | 47.9   | 47.6   | 53.79
 1   | 93.1* | 56.8   | 62.6  | 55.7      | 56.0   | 51.0  | 59.6     | 48.7  | 48.6   | 50.4   | **54.38**
 2   | 87.8*  | 58.2   | 60.2  | 56.1      | 55.7   | 52.6  | 59.0       | 49.3  | 47.9   | 49.9   | 54.32

- **Perturbation value $\kappa$**: In the table below, we can find that the attack success rate will increase with the increase of $\kappa$, because the increase of $\kappa$ will lead to an increase in the degree of image content change. But in terms of image quality, HyperIQA, MUSIQ-Koniq, and TReS degrade image quality as $\kappa$ increases. Since NIMA-AVA and MUSIQ-AVA are trained on AVA, AVA has some post-processed images. When $\kappa$ is small, the effect of ACA can be similar to the post-processing of the image, so these two metrics will increase slightly (For more explanation please refer to **Reviewer oMF6 #Q2**). But as $\kappa$ becomes larger, the degree of image content change becomes larger, and these two metrics also begin to decrease. In summary, we find that $\kappa=0.1$ can achieve a better image quality.

$\kappa$    | Avg. ASR (%) | NIMA-AVA | HyperIQA | MUSIQ-AVA | MUSIQ-Koniq | TReS
:----:|:------------:|:--------:|:--------:|:---------:|:-----------:|:-------:
 0.01 | 28.09        | 5.46     | **0.718**    | 4.31      | **57.85**       | **87.82**
 0.05 | 30.70        | 5.51     | 0.710    | 4.36      | 57.60       | 86.97
 0.1  | 32.10        | **5.54**     | 0.695    | **4.37**      | 56.18       | 85.11
 0.15 | 32.62        | 5.46     | 0.675    | 4.32      | 54.23       | 82.34
 0.2  | **33.39**        | 5.45     | 0.652    | 4.29      | 52.85       | 79.93

---

### Decision · Program_Chairs · 2023-09-21

**Decision:**

Accept (poster)

**Comment:**

In this paper, motivated by the goals of ensuring the photorealism of adversarial examples and boost attack performance, the authors propose a novel unrestricted attack framework called Content-based Unrestricted Adversarial Attack. After the rebuttal process, the responses from the authors mostly satisfy the reviewers and the reviewers reach a consistent consensus of raising the scores and accepting this manuscript. Taking all positive comments from the reviewers into consideration, this paper is recommended to be accepted!